# Label-Free, Rapid and Facile Gold-Nanoparticles-Based Assay as a Potential Spectroscopic Tool for Trastuzumab Quantification

**DOI:** 10.3390/nano11123181

**Published:** 2021-11-24

**Authors:** Ahmed Alsadig, Hendrik Vondracek, Paolo Pengo, Lucia Pasquato, Paola Posocco, Pietro Parisse, Loredana Casalis

**Affiliations:** 1Department of Physics, University of Trieste, 34127 Trieste, Italy; ahmed.mohammed@elettra.eu; 2NanoInnovation Lab, Elettra Sincrotrone Trieste S.C.p.A., 34149 Trieste, Italy; hendrik.vondracek@elettra.eu (H.V.); parisse@iom.cnr.it (P.P.); 3Department of Chemical and Pharmaceutical Sciences, University of Trieste, 34127 Trieste, Italy; ppengo@units.it (P.P.); lpasquato@units.it (L.P.); 4Department of Engineering and Architecture, University of Trieste, 34127 Trieste, Italy; paola.posocco@dia.units.it

**Keywords:** localized surface plasmon resonance (LSPR), functionalized AuNPs, Co(II)NTA, self-assembled monolayer (SAM), HER2/neu receptor, Trastuzumab, biosensing

## Abstract

Monoclonal antibody-based immunotherapy is one of the pillars of cancer treatment. However, for an efficient and personalized approach to the therapy, a quantitative evaluation of the right dose for each patient is required. In this study, we developed a simple, label-free, and rapid approach to quantify Trastuzumab, a humanized IgG1 monoclonal antibody used against human epidermal growth factor receptor 2 (HER2), overexpressed in breast cancer patients, based on localized surface plasmon resonance (LSPR). The central idea of this work was to use gold nanoparticles (AuNPs) as plasmonic scaffolds, decorated with HER2 binders mixed with oligo-ethylene glycol (OEG) molecules, to tune the surface density of the attached macromolecules and to minimize nonspecific binding events. Specifically, we characterized and optimized a self-assembled monolayer of mixed alkylthiols terminated with nitrilotriacetic acid (NTA), and OEG3 as a spacing ligand to achieve both excellent dispersibility and high reliability in protein immobilization. The successful immobilization of histidine-tagged HER2 (His-tagged HER2) on NTA via cobalt (II) chelates was demonstrated, confirming the fully functional attachment of the proteins to the AuNP surface. The proposed design demonstrates the capability of producing a clear readout that enables the transduction of a Trastuzumab/HER2 binding event into optical signals based on the wavelength shifts in LSPR, which allowed for detecting clinically relevant concentrations of Trastuzumab down to 300 ng/mL in the buffer and 2 µg/mL in the diluted serum. This strategy was found to be fast and highly specific to Trastuzumab. These findings make the present platform an auspicious tool for developing affordable bio-nanosensors.

## 1. Introduction

Trastuzumab (commercialized as Herceptin™) is a humanized Immunoglobulin G1 (IgG1) monoclonal antibody that binds to the extracellular domain (ECD) of human epidermal growth factor receptor 2 (HER2) [1]. HER2 or ErbB2 (Neu) belongs to the receptor tyrosine-specific protein kinase family consisting of four epidermal growth factor receptors (EGFRs): (ErbB1), ErbB2 (Neu), ErbB3, and ErbB4 that regulate different biological processes like cell proliferation, migration, and differentiation [2,3]. When normally expressed, the HER2 pathway regulates cell growth and division. HER2 overexpression (up to 100 times), dimerization, and ECD-HER2 shedding have been observed in many types of cancer. Hence, HER2 is considered a cancer biomarker associated with aggressive forms of the disease and poor clinical prognosis [4]. In 1998, the Food and Drug Administration (FDA) approved the use of Trastuzumab in immunotherapy due to its potential inhibitory function to increase the overall survival in HER2-overexpressed metastatic cancers such as gastric and breast cancers [5,6,7]. Although Trastuzumab has improved the clinical outcomes, several studies have reported an increased risk of cardiac failure in patients who received Trastuzumab in combination with chemotherapeutic agents [8,9]. Recent studies have shown that the concentration of Trastuzumab delivered to patients at the beginning of the course of a treatment has a direct impact on the overall survival [10]. It was demonstrated that Trastuzumab did not show the desired clinical benefit in HER2-positive gastric cancer patients when the effective concentration of Trastuzumab was less than 20 µg/mL in the blood plasma after the first administered dose [11]. At the same time, the pharmacokinetics of Trastuzumab is complicated, since the overexpression of HER2 can vary, and the antibody can also bind to the HER2-ECD that is shed from HER2 into the bloodstream.

Up to now, the quantification of Trastuzumab has remained restricted due to the limited capabilities of the available analytical techniques, which are either laborious or not accessible to all laboratories. For instance, B. Cardinali et al. reported the development of a sandwich ELISA protocol to quantify the Trastuzumab concentration in serum using a peptide that mimics the HER2 epitope [12]. The study demonstrated the ability to detect antibody concentrations from 10 to 180 μg/mL in serum. However, despite the appeal, a commercial production of this peptide does not seem within reach. As an alternative to ELISA, high-performance liquid chromatography (HPLC) coupled to tandem mass spectrometry (MS/MS) was described by C.W.N. Damen et al. The presented assay quantified Trastuzumab from 5 to 40 µg/mL in human serum with accuracies <20%, which is complex and requires a particular extraction procedure from human serum [13]. Based on that, the development of a simple, cost-effective, and sensitive biosensor to monitor the functionality and the concentration of Trastuzumab is highly needed.

Biosensors that harness the plasmonic properties of noble metal nano-constructs have fascinated researchers and received considerable attention as alternatives to conventional signal transduction modalities [14,15,16]. Among them, localized surface plasmon resonance (LSPR) sensors, which employ metal nanoparticles (e.g., gold or silver), are increasingly used due to the presence of highly localized electromagnetic fields at the nanoparticle surfaces, which renders them hypersensitive to external environments [17]. Lately, gold nanoparticle (AuNP)-based nanosensor mechanisms are considered a rising hope for fabricating smart sensors [18,19,20]. Since these nanoparticles possess a high surface-to-volume ratio, any tiny perturbation caused within the surroundings can be verified by a spectral shift of their surface plasmonic band. Another advantage of AuNPs is their robust reactivity with thiolated ligands. This allows for surface functionalization that can be specifically geared to detect targets of interest.

The immobilization of biomolecules, including proteins, on 2D or 3D substrates has become a crucial process to develop biosensors [21,22]. While maintaining the functionality of proteins, directed immobilization can be achieved by using affinity-based techniques between a tag and a biocompatible surface. Nitrilotriacetic acid (NTA)-modified materials is a widely adopted technology for the purification and isolation of histidine (His)-tagged recombinant proteins in the presence of Ni^2+^, Cu^2+^, and Co^2+^ ions as chelating agents [23,24,25]. The high affinity of His-tagged protein–NTA interactions and the facile disruption of the complex make NTA highly useful for various applications, including protein purification, labeling, specific immobilization, and the construction of advanced functional materials [26,27,28].

In this work, we aim to explore the use of AuNPs coated with a self-assembled mixed monolayer (mixed SAM) of NTA-modified and oligoethylene glycol (OEG)-modified alkylthios. The novelty of the present work lies in the particle functionalization strategy (Figure 1) that offers: (1) excellent dispersion stability and proper biorecognition endowed by the mixed SAM containing smaller molecules in comparison with bulky PEGylated ligands, (2) the possibility of finely tuning AuNP surface loading to meet the expectation of various bio-applications, and (3) a much faster sample-to-answer LSPR-based plasmonic platform than the lengthy incubation protocols used in ELISA assays. In our design, antibody recognition results in forming a nanoparticle network thanks to the capability of antibodies to link one protein at each arm, leading to plasmonic coupling of the particles that can readily be detected by optical spectroscopy [29,30]. Thus, such an optical readout system could be implemented in affordable point-of-care devices to aid personalized therapies or more sophisticated high-throughput clinical analysis settings. As a proof of principle, in this study, we selected the His-tagged HER2 antigen for the selective detection of Trastuzumab.

## 2. Materials and Methods

### 2.1. Materials

A gold (III) chloride solution, Trisodium citrate tribasic dihydrate, Cobalt(II) chloride hexahydrate, and the standard human serum were purchased from Sigma-Aldrich Chemical Co. (St. Louis, MO, USA). (1-mercaptoundec-11-yl)hexa(ethylene glycol) (HS-(CH_2_)_11_-EG3OH) and (1-mercaptohexadec-16-yl)tri(ethylene glycol) NTA-terminated (for brevity, TOEG3, and NTA, respectively) were purchased from Prochimia Surfaces (Gdynia, Poland). The human HER2/ErbB2 protein, His-tagged, was purchased from Acro Biosystems (Newark, DE, USA). The monoclonal antibody Trastuzumab was purchased from MedChemExpress (MCE^®^, Monmouth Junction, NJ, USA). The HEPES buffer used in this study was made of 100-mM HEPES buffer containing 0.025% Tween 20, pH 7.4 prepared in ultrapure water (Milli-Q water). Severe acute respiratory syndrome-2019 novel coronavirus (SARS-CoV-2(2019-Ncov)) spike neutralizing antibody, Anti- RBD Rabbit mAb, was purchased from Sino Biological (Beijing, China) and used as a negative control for the specificity test. 

### 2.2. Synthesis, Characterization, and Functionalization of AuNPs

AuNPs of approximately 13 nm a diameter were synthesized by a sodium citrate reduction of hydrogen tetrachloroaurate (HAuCl_4_) as described elsewhere [31]. Briefly, 1.1 mL of 17.3-mM HAuCl_4_ was added to 43.6 mL of Milli-Q water and brought to a boil with vigorous stirring on a magnetic stirrer hotplate. Then, 300 μL of 72 mg/mL of trisodium citrate dihydrate was added rapidly. The solution underwent a color change from yellow to colorless, then dark blue and, finally, deep red within a few minutes. The reaction was maintained at the boiling point for 30 min to assure the complete reduction of the gold salt; after which, the solution was cooled down to room temperature. The resulting gold colloids were protected from light and stored at 4 °C until required. The final concentration of the colloid solution was 3.94 nM. The optical properties, the morphology, and the hydrodynamic diameter of the AuNPs were verified by Ultraviolet–Visible (UV–Vis) spectroscopy (Perkin-Elmer lambda 25, Waltham, MA, USA), transmission electron microscopy (TEM) (Philips, Eindhoven, The Netherlands), and dynamic light scattering (DLS) (Malvern Instrument, UK), respectively. The functionalization of AuNPs with NTA and TOEG3-mixed SAM was obtained via a place exchange approach. After determining the minimum concentration of TOEG3 that fully covers the AuNPs, the citrate ligands covering the AuNPs were replaced by NTA and TOEG3 ligands gently mixed at a 50:50 ratio, forming a fully mixed monolayer on the AuNPs by self-assembling overnight at room temperature. Following the protocol described in Reference [19], the particles were purified via centrifugation (2×, 7200× *g* for 45 min). The final pellet was redispersed in 500 µL of the buffer (100-mM HEPES containing 0.025% Tween 20, pH 7.4). Successful surface functionalization was confirmed by monitoring the change of the hydrodynamic diameter of particles using DLS, gel electrophoresis, and the characteristic surface plasmon band (SPB) peak before and after the passivation using UV–Vis spectroscopy.

### 2.3. Incorporation of Co^2+^ Cations and Immobilization of Histidine-Tagged HER2 Protein

Functionalized AuNPs (450 µL, 2 nM) were incubated with CoCl_2_ (50 µL, 100 mM) for 1 h at room temperature. After charging with Co^2+^, the particles were centrifuged (2×, 7200× *g* for 30 min) and washed twice with the buffer. Ultimately, the particles were suspended in the same buffer and filtered through a 0.1-μm syringe filter. For the protein immobilization onto the particles, 2 µL, 6.3 µM of His-tagged HER2 protein (His-tagged HER2, 0.1% BSA in PBS) was introduced to the mixture and left to react at room temperature under mild shaking for 1 h. Following that, the particles were centrifuged (2×, 5000× *g* for 20 min) and, finally, redispersed in 500 µL of the buffer. Subsequently, the particles were characterized using UV–Vis spectroscopy and DLS.

### 2.4. LSPR-Based Detection of Trastuzumab using His-Tagged HER2 Decorated AuNPs

Trastuzumab antibody at various concentrations ranging from 0.1 to 40 µg/mL was added to His-tagged HER2/AuNP bioconjugates in a final volume of 500 μL, well-redispersed, and left to react for 5 min before conducting the spectroscopic scan. The corresponding absorption spectra were recorded in the wavelength range from 400 to 750 nm. The degree of aggregation was assessed by computing the ratio of the absorption recorded at two selected wavelengths (A_600_/A_520_) and normalizing this ratio to the corresponding values before addition of the antibody.

## 3. Results and Discussion

### 3.1. Characterization of Synthesized AuNPs

Most commonly, AuNPs are synthesized using the reduction of the soluble gold salt (HAuCl_4_) in the presence of capping agent such as sodium citrate, a compound able to attach to the nanoparticle surface, preventing its growth beyond the desired size and conferring stability of the colloid in the solvent used. The particle shape and dimensions can be controlled by tuning the experimental parameters, such as the reaction time, temperature, and most importantly, the ratio between the reducing agent used and gold precursor. The UV–Vis spectrum, as shown in Appendix A, of AuNPs synthesized using the citrate reduction method and the corresponding TEM of citrate-capped AuNPs exhibited a maximum absorption at 518 nm, consistent with the typical SPR band of AuNPs. This renders a colloidal solution of spherical AuNPs, which is visibly red in color. The TEM in Appendix A revealed a size distribution of the resulting nanoparticles of 13 ± 1 nm, while the DLS measurements showed that the particles have an average diameter of 15 ± 6 nm, which is compatible with the AuNP size considering the hydration shell. The selected diameter size (13 nm) was based on finding a compromise between having a significant loading of biorecognition sites while maintaining the advantages of the small size.

### 3.2. Passivation of AuNPs with NTA/TOEG3 SAM

The variety of applications of NTA-based technology in the purification, immobilization, and separation of His-tagged macromolecules is unparalleled. The incorporation of the NTA moiety on nanosized particles was reported by Xu et al., demonstrating the ability of these nanoparticles to chelate to bidentate Ni^2+^ or Co^2+^ cations [32]. Later, Chen et al. further employed this strategy by incorporating Ni-NTA onto superparamagnetic particles to enrich His-tagged proteins and other phosphorylated peptides [33]. Sosibo et al. reported the synthesis of stable, hydrophilic monolayer-protected clusters of gold (Au-MPCs) functionalized with PEG-NTA and co-stabilized with PEG-OH as probes for targeting histidine-tagged proteins [34]. It was demonstrated that it is crucial to incorporate a spacing ligand, forming a mixed monolayer onto the particle surface to enhance the long-term stability of NTA-functionalized nanoparticles. However, long PEGylated alkylthiol might hinder the plasmonic effect we would like to exploit for high sensitivity biorecognition. In the present study, in a single-step reaction, the shorter NTA/TOEG3 were made self-assembling onto AuNPs via the ligand place exchange approach of the loosely bound citrates capping the colloidal surface. TOEG3 was chosen as the spacing ligand due to its high hydrophilicity that promotes favorable interactions with the gold colloids. Besides, TOEG3 acts as a protein-repellent thiol, reducing the risk of the nonspecific adsorption of biomolecules onto AuNP surfaces, as already demonstrated in our group on flat Au surfaces [35]. In previous studies on AuNPs, however, it was reported that short-chain OEG thiols with ≤4 EG units can stabilize only a suspension of small AuNPs (<3 nm). For larger AuNPs (>10 nm), OEG thiols <6 EG units were found to be insufficient to stabilize the colloids [36]. Contrastingly, from our UV–Vis absorption measurements reported in Figure 2a, we found that, starting from the concentration of 5 µM, TOEG3 successfully prevented a 2-nM solution of 13-nm diameter AuNPs from aggregation. In this condition, we assumed a fully covered SAM was formed on the nanoparticle. As a starting point for the formation of mixed, functional SAM, we then used a 50:50 concentration of NTA and TOEG3-terminated alkylthiols and 2.5 µM of NTA plus 2.5 µM of TOEG3 to form a homogenous and stable mixed SAM onto the AuNP surface. The successful ligands attachment was confirmed by the 3-nm redshift in the UV–Vis spectrum, as shown in Figure 2b. Gel electrophoresis revealed a slower mobility of NTA/TOEG3 than only NTA (see Figure 2b for the discrepancy)-coated particles, indicating surface modification of the particles towards the formation of a mixed monolayer of the dissimilar ligands that are covalently bonded to AuNP surfaces. This finding also demonstrates the potential of agarose gel electrophoresis to confirm the SAM formation of charged SAMs on the surfaces of the AuNPs.

### 3.3. Incorporation of Co^2+^ ations and Immobilization of the His-Tagged HER2 Protein

To detect Trastuzumab, the His-tagged HER2 protein was first anchored on the AuNPs surface based on the Co(II)–NTA complexation strategy depicted in Figure 1. The quadridentate NTA ligand occupies four coordination positions on the hexadentate central cobalt cation, allowing for the availability of the two vacant binding sites for the His-tagged biomolecules to attach. Following the reaction of the modified AuNPs with excess CoCl_2_, there was no observable change in the optical properties of the particles, as verified by UV–Vis spectroscopy. It is noteworthy that controlling the reaction time and the concentration of CoCl_2_ is crucial to prevent any agglomeration of the particles. When the Co^2+^ concentration is high, particles form visible aggregation during the purification process, likely because of the formation of polynuclear Co–NTA complexes instead of well-defined mononuclear ones that may trigger NP aggregation, while, if the Co^2+^ concentration is too low, there are not enough ions on the particles to conjugate enough proteins on the particle surface. However, in our study, incubating with 10 mM of CoCl_2_ for 1 h was ideal to charge the particles without causing any agglomeration. Upon the introduction of the His-tagged HER2 protein, the resultant bioconjugates were readily soluble in the solvent buffer and showed the typical red color of the gold colloids. While the UV–Vis absorption spectra remained identical to that of the NTA/TOEG3-modified particles spectrum, the DLS readouts confirmed the successful functionalization upon each step of preparation (Figure 3). This is not surprising, given the different principles of both techniques. Whereas UV–Vis spectrophotometry relies on detecting changes in the optical properties reflected in LSPR, DLS can reveal differences in the particle hydrodynamic diameter size upon functionalization that do not necessarily cause observable variations in the overall optical properties.

### 3.4. Detection of Trastuzumab by LSPR in Buffer

After demonstrating that Trastuzumab binds to the His-tagged HER2 protein receptor using indirect ELISA (details in Appendix A), the sensing capability of our platform was then tested by monitoring the LSPR shifts induced by the immunoconjugate recognition across a wide range of Trastuzumab concentrations (0.1–40 µg/mL), physiologically relevant in anti-HER2 cancer treatment, in 100-mM HEPES buffer containing 0.025% Tween 20 (pH 7.4). The range tested here covers the therapeutic relevant range of 5–40 µg/mL, i.e., the range of concentrations of circulating Trastuzumab in the blood used to dose the therapy in a personalized manner [37]. The LSPR peak of AuNP depends on the geometry, surface chemistry, and the refractive index of the surrounding environment [38]. Nanomaterials that exhibit high refractive index sensitivities are strongly desired for fabricating plasmonic nanosensors. Herein, it was expected that shifts in LSPR would be dependent on the amount of antibody bound to antigen-decorated particles. As clearly observable from Appendix A, the LSPR undergoes a redshift upon the addition of a concentration of at least 1.3 µg/mL of Trastuzumab. Increasing concentrations of the antibody led to a more substantial redshift of the LSPR band due to the binding of the immunoconjugate. A similar trend was also observed in previous studies [39,40,41]. Generally, the LSPR spectral shifts are attributed to the combination of local refractive index changes induced by analyte adsorbed onto the particle surface and the coupling of surface plasmons of adjacent particles. The latter heavily depends on multiple interactions between the ligands and target receptors. In the present case, the significant redshift presumably corresponds to the interparticle coupling of surface plasmons due to the presence of Trastuzumab. It shall be mentioned that the absorbance measurements were taken 5 min after completing the titration process for the respective amount of the antibody. The timeframe required in this step of biorecognition, especially in comparison to the lengthy incubation protocols used in ELISA assays, highlights the utility of this platform for the development of a rapid LSPR-based plasmonic nanosensor. The shift in LSPR as a function of Trastuzumab concentration was quantified as shown in Figure 4 (blue dots). Additionally, the normalized ratio of the absorbance at 600 nm (A_600_) and at 520 nm (A_520_) with respect to the antibody concentration was determined. This relation reveals the arrangement of the particles in the assay suspension. Whereas the former is related to the wavelength at which aggregation is observed, the latter corresponds to the LSPR absorption peak of unperturbed AuNPs. Normalization to the absorbance value displayed by the nanoparticles decorated with His-tagged HER2 enables us to quantify NP clusterization: the higher the ratio, the greater the degree of aggregation. The normalized ratio of the aggregation was plotted as a function of the target antibody concentration. Figure 4 (red dots) shows a near-sigmoidal titration curve for the various concentrations used in this study. When a small amount of antibody was added (300 ng/mL), the aggregation induced resulted in a small rise in the aggregation ratio (>1), presumably due to the onset formation of antibody-mediated dimers. The presence of an increasing quantity of the antibody promotes the aggregation with a monotonic increase of the signal observed between 4 µg/mL and 20 µg/mL before reaching a plateau at a concentration of 40 µg/mL of the target analyte. First, we can conclude that the limit of Trastuzumab detection (LOD) of our assay is as low as 300 ng/mL by examining the formation assemblies through the aggregation ratio relationship, which gives a more sensitive and a better quantitative description of the particle status with increasing concentrations of the target analyte. Moreover, the saturation is reached at 40 µg/mL, which is the limit of the clinically relevant concentrations of antibody freely circulating in the blood in Trastuzumab-based therapy. To further validate this result, TEM micrographs were acquired, shown in Figure 5a–d, displaying the morphology of the His-tagged HER2-decorated particles before and after the addition of Trastuzumab (2, 10, and 20 µg/mL) (see, also, Appendix A). The findings demonstrate that the capture of antibodies leads to a decrease in the interparticle distance and, subsequently, leads to cluster formation, enabling us to develop a rapid sensing platform using a simple UV–Vis spectrophotometer. The DLS readouts further confirmed our observations by detecting the clusterization of AuNPs induced by the Trastuzumab in the assay solution. As shown in Figure 5e, the average hydrodynamic diameter of His-tagged HER2-decorated AuNPs was greatly increased from 37 nm to 265, 528, 747, 837, and 923 nm upon binding to Trastuzumab with concentrations ranging from 2 to 10 µg/mL, respectively, indicating that the modified AuNPs were agglutinated together into aggregates.

### 3.5. Evaluating the Specificity of the Platform

Having tested the concentration-dependent LSPR shifts of His-tagged HER2-decorated particles over a wide range of Trastuzumab concentrations, the next step was to evaluate the specificity of the platform. In this case, we used a non-HER2-specific antibody, Anti-RBD, to interact with the particles instead of Trastuzumab, using the same method to observe if any assembly would form. As expected, upon incubation with various concentrations of off-target antibodies, no change in the UV–Vis absorption spectra was observed (Appendix A), indicating that the particles sensed no or negligible interactions. These findings demonstrate the high specificity of our platform for detecting Trastuzumab.

### 3.6. Elucidating the Impact of Free His-Tagged HER2 Proteins on LSPR Response

To acquire a deeper understanding about the role played by the free His-tagged HER2 proteins in the LSPR response, we devised an experiment in which a dispersion of the bioconjugates was treated with increasing amounts of free His-tagged HER2 prior to the addition of a fixed concentration of Trastuzumab (10 µg/mL). As can clearly be seen in Appendix A**,** an increasing amount of the proteins caused a dramatic drop in the normalized ratio (A_600_/A_520_), indicating a marginal aggregation of the nanoparticles. Interestingly, nearly 5 nM of free HER2 protein in the colloidal suspension was sufficient to cause a drastic drop in the normalized ratio (A_600_/A_520_). This implies that all antigen-binding recognition sites of the antibody were saturated and no longer accessible for bridging His-tagged HER2 immobilized on decorated AuNPs. Thus, our analysis indicates the importance of removing unbound proteins after the conjugation process with Co(II) NTA.

### 3.7. Evaluating the LSPR Response upon Tuning the Density of the Surface Grafted SAM

To investigate the possibility of optimizing the density of the His-tagged HER2 protein immobilized onto the particle surface, we varied the molar ratio of the NTA anchor and the TOEG3 spacer ligands up to the saturation density. The coating densities were adjusted by using the 20:80, 50:50, and 80:20 ratios between NTA and TOEG3, respectively. At first glance, Figure 6 shows that the aggregation ratio measured with UV–Vis gradually increases for the various AuNP–Trastuzumab immunocomplexes when the amount of NTA–ligand increases. AuNPs passivated with increasing NTA-coating density also revealed a gradual increase in the hydrodynamic size, indicating a higher protein loading capacity. However, we observed that, with a 80:20 coverage density, the chance of having aggregates after protein loading is relatively high (see Appendix A), likely because there are still Co^2+^ ions bound to the NP surface. Thus, the chance for electrostatic interactions with other NTA-terminated AuNPs is increased. Therefore, we maintained a 50:50 ratio for the rest of the study.

### 3.8. Evaluating Assessment of Trastuzumab detection by LSPR in Human serum

At this stage, we investigated the biosensor performance in detecting trastuzumab in a complex matrix, such as human serum. Solutions containing His-tagged HER2-decorated particles were incubated with a range of diluted human serum (100, 75, 50, 25, and 10×) containing Trastuzumab (2, 10, and 20 µg/mL) to show the results of direct detection in diluted human serum. As seen in Figure 7, it is highly challenging to detect Trastuzumab in a colloidal suspension containing tenfold diluted serum. This is not surprising, as, in extremely complex biological mixtures, several proteins, including transferrin (TF) or histidine-rich glycoprotein (HRG), are present in high concentrations and can act as interferents [19]. It was reported that the latter highly interacts with several ligands, including metal ions [23]. To verify this possibility, we incubated the bioconjugates in 10% serum to allow the possible displacement of the His-tagged HER2 protein from the surface. Then, the bioconjugates were separated from the serum and evaluated in an assay with various concentrations of Trastuzumab (in buffer). A shown in Appendix A, no LSPR response was observed across all the doses of Trastuzumab, demonstrating a loss of sensitivity due to less effective conjugates. The presence of histidine-rich proteins that may highly interfere with the conjugation of the immunocomplex was investigated by employing Ni-NTA agarose beads (details in S1.4). Contrastingly, upon the purification trial conducted to remove proteins containing histidine repeats in 1% and 5% diluted serums (Appendix A), the shifts in LSPR became more pronounced with the addition of 2 µg/mL of the target antibody, as seen in Appendix A. Additionally, as reported in the same figure, the gel electrophoresis showed a slower mobility for the immunocomplex dispersed in the post-treated 5% diluted serum, indicating a higher chance for Trastuzumab to recognize His-tagged HER2-decorated particles compared to nontreated serum. These findings are in agreement with the findings of a recent report that demonstrated that 100 times the dilution of the human serum was the optimum for detection using NPs [42]. However, further optimization and the investigation of other possible routes to filter the biological media in order to enhance the sensitivity of the proposed platform are part of ongoing research.

## 4. Conclusions

Optical, label-free biosensors represent precious means for real-time, direct analyses of biomolecular interactions that can be exploited to boost the development of precise and individualized therapies for severe diseases like cancer. In this work, we described the development of a facile, rapid, and specific AuNP-based platform for the detection and quantification of Trastuzumab. To accomplish this, the His-tagged HER2 protein was anchored on AuNP surfaces by employing the coordination chemistry of metal complexes attached to a mixed self-assembled monolayer (mixed SAM) onto the nanoparticle surface. We characterized and optimized the Co(II)NTA AuNP platform incorporated with TOEG3 as a spacing ligand for passivating the particle surface to achieve both excellent dispersibility and high reliability in protein immobilization. The successful tethering of the His-tagged HER2 protein was demonstrated, confirming the fully functional attachment of the protein to the Co(II)NTA/TOEG3@AuNP platform. This methodology is readily applicable to other histidine-rich biomolecules. In line with the expectations for this type of essay, the biorecognition of antibodies and antigen-decorated AuNP conjugates results in a substantial LSPR shift due to the formation of particle assemblies, enabling the detection of the antibody in clinically relevant concentrations. Although our sensing platform showed a different sensitivity when comparing the buffer and the human serum matrix, with further optimizations and improvements, we anticipate our platform will pave the way for fabricating lab-free and ready-to-use nanoprobes, which may have application potentials in point-of-care medical diagnoses.

## Figures and Tables

**Figure 1 nanomaterials-11-03181-f001:**
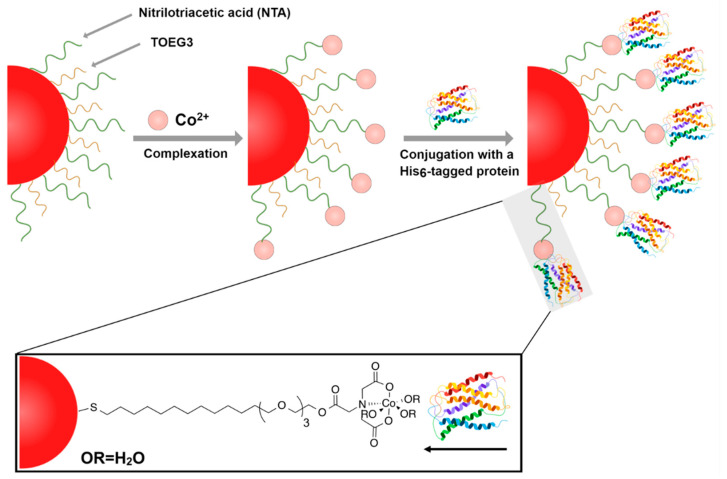
A representation illustrates the attachment strategy of the His-tagged protein onto the surface of the Co(II)NTA/TOEG3 AuNPs.

**Figure 2 nanomaterials-11-03181-f002:**
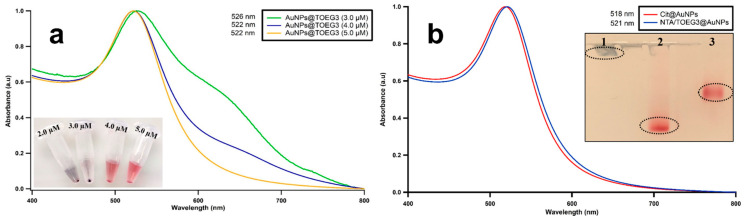
(**a**) UV–Vis absorption spectra of AuNPs coupled with different concentrations of TOEG3. The blue color indicated that the TOEG3 was insufficient to cover the surfaces of particles. Once a sufficient amount was used, the colloidal solution does not form agglomerates, and the red color is maintained. (**b**) UV–Vis spectra of citrate and SAM-protected AuNPs to monitor the surface modification. Inset: Gel electrophoresis bands for: (1) citrate-capped AuNPs, (2) NTA@AuNPs, and (3) NTA/TOEG3@AuNPs.

**Figure 3 nanomaterials-11-03181-f003:**
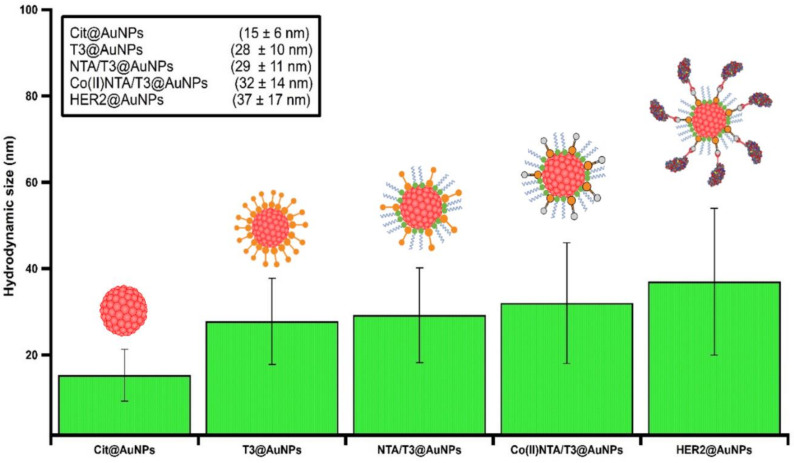
Volume-weighted size distribution DLS measurements of AuNPs after different steps of the preparation.

**Figure 4 nanomaterials-11-03181-f004:**
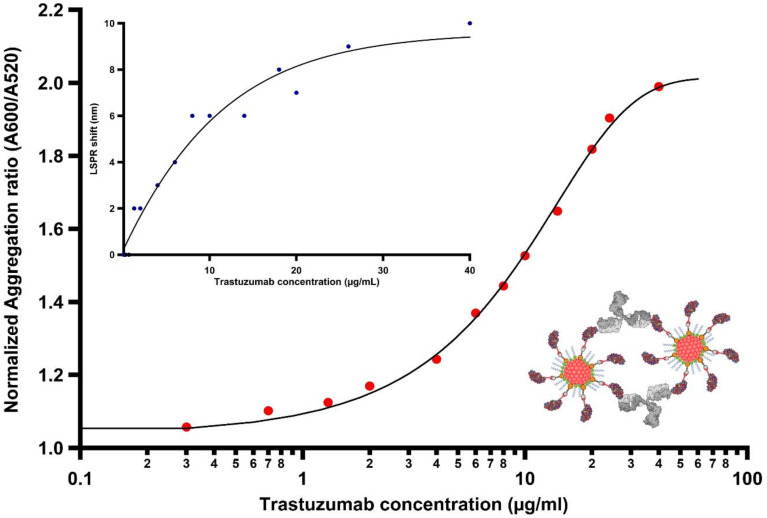
Quantification of the LSPR shift (blue dots) and normalized aggregation ratio (red dots) of immunoconjugates with various concentrations of Trastuzumab. Dose-response presented in each plot was fitted with the four-parameter logistic sigmoidal curve.

**Figure 5 nanomaterials-11-03181-f005:**
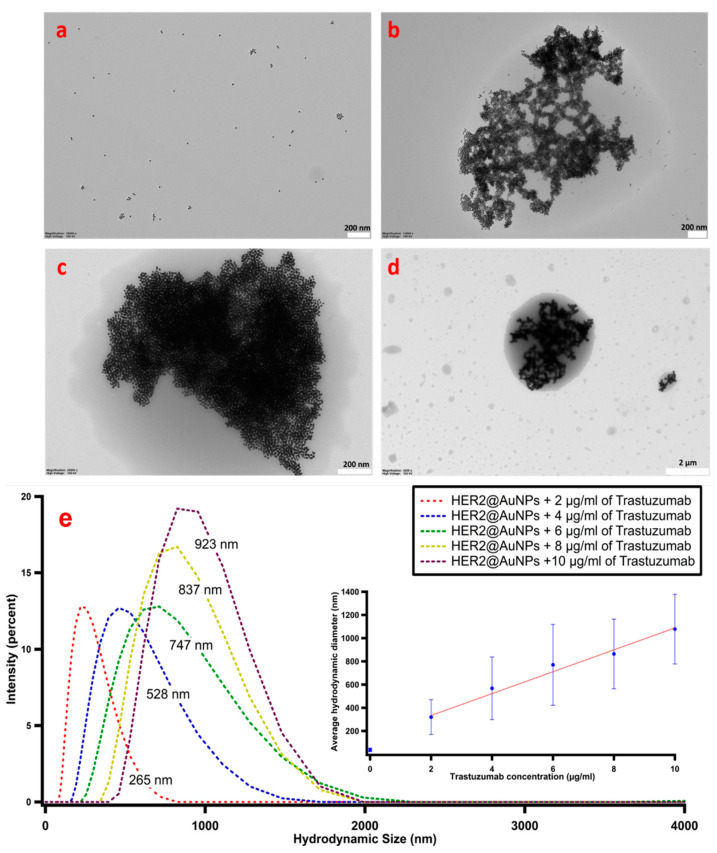
(**a**–**d**) Representative TEM images of (**a**) HER2-decorated AuNPs (**b**–**d**) incubated with 2, 10, and 20 µg/mL of Trastuzumab. The formation of a 3D network indicates the successful coupling of AuNPs through Trastuzumab linking. (**e**) DLS readouts for AuNPs probes treated with various concentrations of Trastuzumab. Upon mixing the HER2-decorated AuNP probes with Trastuzumab, the particles are clustered together, leading to an average particle size increase of the assay solution.

**Figure 6 nanomaterials-11-03181-f006:**
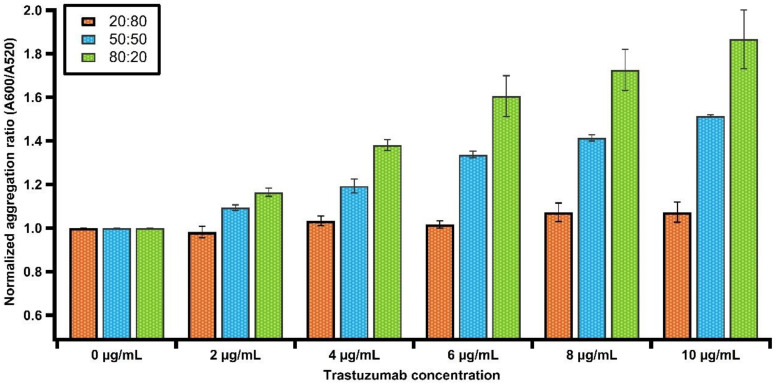
Comparison of the LSPR response for various concentrations of Trastuzumab upon adjusting the SAM density on the surfaces of AuNPs. Note that, with a 20:80 ratio, no plasmonic response was observed across all doses of the antibody.

**Figure 7 nanomaterials-11-03181-f007:**
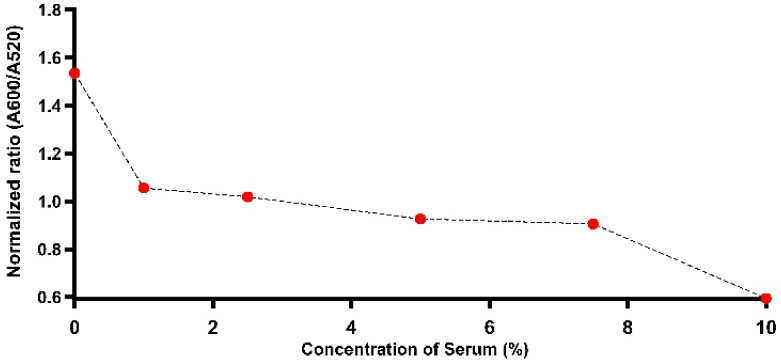
Plasmonic response for His-tagged HER2-decorated particles incubated with various dilutions of human serum-containing Trastuzumab.

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
