# Peer review of "Label-Free, Rapid and Facile Gold-Nanoparticles-Based Assay as a Potential Spectroscopic Tool for Trastuzumab Quantification"

_nanomaterials, 2021, doi:10.3390/nano11123181_

Round 1

Reviewer 1 Report

Excellent study, nice to see that someone knows how to properly interpret DLS data! Very few, very minor suggestions:

Line # 59 and Line # 127, change capital letter to lower case of first word of the technique "Localized" to "localized"", "Gel" to "gel".

reference #15 needs amending, line breaks have wreaked havoc with the numbering.

Author Response

We are grateful to the Reviewer for the enthusiastic comment. We corrected the manuscript including all the Reviewer’s suggestions. Comments in the text are highlighted in red.

Reviewer 2 Report

The major concern of aggregation-based assay is that the aggregation will take place only at specific sensor:target ratios. At low sensor-target ratios, no aggregation would occur. As the sensor-target ratio increases, the number and size of aggregates also gradually increase. However, when passing beyond the optimal ratio, the sensor surfaces become saturated with bound targets. In the presence of excess targets, aggregation of sensor molecules and targets will no longer occur. The aggregation curve follows the bell-shaped distribution. An example can be seen at https://courses.lumenlearning.com/microbiology/chapter/detecting-antigen-antibody-complexes/.

Figures 6, 7, 8, 12, and 14 should be replotted to include trastuzumab concentrations beyond the concentrations evaluated to determine the maximum detection limit and to acknowledge, e.g., that at a given hydrodynamic diameter measures, it could represent two different estimated trastuzumab concentrations.

Author Response

We thank the Reviewer for the valuable comments. We point out here that in the range of Trastuzumab concentration of clinical relevance (few- 40 mg/mL), the one tested in our study, we are clearly working below saturation in our plasmonic aggregation assay. It might happen that beyond 40 mg/mL we enter a saturated regime as explained by the Reviewer. However, testing a value of Trastuzumab concentration higher than 40 mg/mL is very costly, and is not clinically relevant. We made this point clearer in the text (lines 317-319). Moreover we point out here that: 1) Trastuzumab is a monoclonal antibody, while the precipitin assay cited by the Reviewer uses polyclonal antibodies, which recognize multiple epitopes of the same antigen, favouring visible aggregates formation; 2) we optimized the number of HER2 molecules per AuNPs, in order to increase the sensitivity of our plasmonic assay in the range of concentrations of the antibody requested. We hope this helped to better clarify the validity of the proposed assay.

Reviewer 3 Report

Dear authors,

The paper presents the "Label-free, Rapid and Facile Gold-Nanoparticles-Based Assay as a Potential Spectroscopic Tool for Trastuzumab Quantification" submitted to the Nanomaterials (Manuscript Number nanomaterials-1433893). In my opinion, the manuscript is suitable for publication in the journal after the following points are addressed:

- Please, double-check if the best results obtained by the authors have been described in the Abstract;

- I think the Objective of the work can be improved, showing what the group intends to present in the work regarding what does not yet exist in the literature;

- Please, double-check the abbreviations used in the text, in the first mention of them, provide the meaning, followed by the abbreviations;

- Please, change "hour" to "h", "ml" to "mL", "μl" to "μL", and "5. Conclusions" to "4. Conclusions";

- Please, double-check the Material and Methods section, I think some methodologies employed were based on published works, so in this case, should be cited in the text;

- The discussion (3. Results and Discussion section) is merely described and is not discussed or correlated with data obtained by other research groups. For me, this section should be rewritten and mainly better addressed by the authors;

- There are many Figures in the manuscript. Please, include some of them in the Supplementary Material.

Author Response

We thank the Reviewer for the noteworthy recommendations. Those helped us to present the results in a much clearer way. We carefully addressed all the points raised.

- Please, double-check if the best results obtained by the authors have been described in the Abstract

We modified the text of the abstract to highlight all the achievements and novelty of the assay

- I think the Objective of the work can be improved, showing what the group intends to present in the work regarding what does not yet exist in the literature;

We thank the Reviewer for the comment.  We better clarified what was already present in the literature and what is the scope of our research.

- Please, double-check the abbreviations used in the text, in the first mention of them, provide the meaning, followed by the abbreviations;

We acted as recommended. Text was amended accordingly

- Please, change "hour" to "h", "ml" to "mL", "μl" to "μL", and "5. Conclusions" to "4. Conclusions";

We amended the text as recommended

- Please, double-check the Material and Methods section, I think some methodologies employed were based on published works, so in this case, should be cited in the text;

We added few references to the text in the Materials and Methods section, as suggested

- The discussion (3. Results and Discussion section) is merely described and is not discussed or correlated with data obtained by other research groups. For me, this section should be rewritten and mainly better addressed by the authors;

We rewrote the section as recommended. We improved the discussion of the results

- There are many Figures in the manuscript. Please, include some of them in the Supplementary Material

The Reviewer, also in this case, is absolutely right. We moved former Figs. 1, 6, 10, 11 and 14 to the SI, and grouped former Figs. 2 and 3 (now Fig. 2) and Figs. 8 and 9 (now Fig. 5).

All the changes have been highlighted in red in the new version of the manuscript.

Round 2

Reviewer 2 Report

The authors have properly addressed the reviewers' comments and suggestions. The revised manuscript is suitable for publication.

There is a typo in Line 200. "PEGilated alkylthiol" should be spelled as "PEGylated alkylthiol"

Reviewer 3 Report

Dear authors,

The paper presents the "Label-free, Rapid and Facile Gold-Nanoparticles-Based Assay as a Potential Spectroscopic Tool for Trastuzumab Quantification" submitted to Nanomaterials (Manuscript #nanomaterials-1433893). After the revision performed by the authors, I recommend the manuscript for publication.